# Therapeutic Potential of a Novel Vitamin D_3_ Oxime Analogue, VD1-6, with CYP24A1 Enzyme Inhibitory Activity and Negligible Vitamin D Receptor Binding

**DOI:** 10.3390/biom12070960

**Published:** 2022-07-08

**Authors:** Ali K. Alshabrawy, Yingjie Cui, Cyan Sylvester, Dongqing Yang, Emilio S. Petito, Kate R. Barratt, Rebecca K. Sawyer, Jessica K. Heatlie, Ruhi Polara, Matthew J. Sykes, Gerald J. Atkins, Shane M. Hickey, Michael D. Wiese, Andrea M. Stringer, Zhaopeng Liu, Paul H. Anderson

**Affiliations:** 1UniSA Clinical and Health Sciences, Health and Biomedical Innovation, University of South Australia, Adelaide, SA 5001, Australia; ali.alshabrawy@mymail.unisa.edu.au (A.K.A.); cyan.sylvester@mymail.unisa.edu.au (C.S.); emilio.petito@mymail.unisa.edu.au (E.S.P.); kate.barratt@unisa.edu.au (K.R.B.); rebecca.sawyer@unisa.edu.au (R.K.S.); jessica_kate.heatlie@mymail.unisa.edu.au (J.K.H.); polra001@mymail.unisa.edu.au (R.P.); matt.sykes@unisa.edu.au (M.J.S.); shane.hickey@unisa.edu.au (S.M.H.); michael.wiese@unisa.edu.au (M.D.W.); andrea.stringer@unisa.edu.au (A.M.S.); 2Pharmaceutical Chemistry Department, Faculty of Pharmacy, Helwan University, Cairo 11795, Egypt; 3Department of Medicinal Chemistry, Key Laboratory of Chemical Biology (Ministry of Education), School of Pharmaceutical Sciences, Cheeloo College of Medicine, Shandong University, Jinan 250012, China; yingjiecui@sina.cn (Y.C.); liuzhaop@sdu.edu.cn (Z.L.); 4Department of Pharmacy, Shandong Provincial Hospital Affiliated to Shandong First Medical University, Jinan 250021, China; 5Centre for Orthopaedic and Trauma Research, Faculty of Health and Medical Sciences, The University of Adelaide, Adelaide, SA 5005, Australia; dongqing.yang@adelaide.edu.au (D.Y.); gerald.atkins@adelaide.edu.au (G.J.A.)

**Keywords:** vitamin D_3_, CYP24A1, in silico docking, catabolism inhibition, HEK293T

## Abstract

The regulation of vitamin D_3_ actions in humans occurs mainly through the Cytochrome P450 24-hydroxylase (CYP24A1) enzyme activity. CYP24A1 hydroxylates both 25-hydroxycholecalciferol (25(OH)D_3_) and 1,25-dihydroxycholecalciferol (1,25(OH)_2_D_3_), which is the first step of vitamin D catabolism. An abnormal status of the upregulation of CYP24A1 occurs in many diseases, including chronic kidney disease (CKD). CYP24A1 upregulation in CKD and diminished activation of vitamin D_3_ contribute to secondary hyperparathyroidism (SHPT), progressive bone deterioration, and soft tissue and cardiovascular calcification. Previous studies have indicated that CYP24A1 inhibition may be an effective strategy to increase endogenous vitamin D activity and decrease SHPT. This study has designed and synthesized a novel C-24 *O*-methyloxime analogue of vitamin D_3_ (**VD1-6**) to have specific CYP24A1 inhibitory properties. **VD1-6** did not bind to the vitamin D receptor (VDR) in concentrations up to 10^−7^ M, assessed by a VDR binding assay. The absence of VDR binding by **VD1-6** was confirmed in human embryonic kidney HEK293T cultures through the lack of *CYP24A1* induction. However, in silico docking experiments demonstrated that **VD1-6** was predicted to have superior binding to CYP24A1, when compared to that of 1,25(OH)_2_D_3_. The inhibition of CYP24A1 by **VD1-6** was also evident by the synergistic potentiation of 1,25(OH)_2_D_3_-mediated transcription and reduced 1,25(OH)_2_D_3_ catabolism over 24 h. A further indication of CYP24A1 inhibition by **VD1-6** was the reduced accumulation of the 24,25(OH)D_3_, the first metabolite of 25(OH)D catabolism by CYP24A1. Our findings suggest the potent CYP24A1 inhibitory properties of **VD1-6** and its potential for testing as an alternative therapeutic candidate for treating SHPT.

## 1. Introduction

The 25-hydroxyvitamin D 24-hydroxylase (CYP24A1) enzyme is a mitochondrial inner membrane cytochrome P450 component that acts naturally to catabolize both 25-hydroxycholecalciferol (25(OH)D_3_) and 1,25-dihydroxycholecalciferol (1,25(OH)_2_D_3_) to control vitamin D_3_ hormonal actions in different tissues [1]. This catabolism occurs via hydroxylation at either C23 or C24 of the aliphatic side chain of 25(OH)D_3_ and 1,25(OH)_2_D_3_, resulting in less-active catabolites and culminating in the formation of calcitroic acid [2]. CYP24A1 gene expression in healthy individuals is regulated by the effects of 1,25(OH)_2_D_3_, parathyroid hormone (PTH), and fibroblast growth factor 23 (FGF23) [3], with 1,25(OH)_2_D_3_ being the predominant stimulator of expression [4]. The control of PTH over CYP24A1 expression was found to be tissue-dependent, with PTH increasing CYP24A1 activity in bone cells through enhancing 1,25(OH)_2_D_3_ mediated induction [5], while directly suppressing CYP24A1 activity in the kidneys through a non-vitamin D receptor (VDR)-dependent pathway [6]. FGF23, on the other hand, stimulates CYP24A1 expression in kidney cells to lower 1,25(OH)_2_D_3_-mediated phosphate absorption [7].

Chronic kidney disease (CKD) is a progressive disease characterized by the gradual deterioration of kidney function. CKD is a growing public health issue, mainly due to aging, with comorbidities such as hypertension and diabetes being major contributors [8,9]. An inevitable event in CKD is the disturbance of the renal vitamin D metabolic pathways, which involve 25-hydroxyvitamin D 1α-hydroxylase (CYP27B1) and CYP24A1. The gradual reduction in renal function translates into a progressive decline in CYP27B1 activity, leading to a decrease in the production of circulating 1,25(OH)_2_D_3_ [2]. Additionally, elevated serum FGF23 levels are one of the first indicators of renal disease, which continues to rise as CKD progresses due to hyperphosphatemia resulting from impaired renal phosphate excretion [10,11,12]. Elevated FGF23 suppresses renal CYP27B1 and stimulates renal CYP24A1 expression, contributing to inadequate 1,25(OH)_2_D_3_ production [13]. Circulating levels of 25(OH)D_3_, as a precursor substrate for 1,25(OH)_2_D_3_, have also been reported to decline in CKD due to factors including proteinuria [14]. Inappropriately low levels of 1,25(OH)_2_D_3_ lead to impaired intestinal calcium absorption and hypocalcaemia, which in turn induce secondary hyperparathyroidism (SHPT) [15,16].

To reduce PTH levels, 1,25(OH)_2_D_3_ or VDR activating vitamin D analogues, such as alfacalcidol, maxacalcitol, doxercalciferol, or paricalcitol, are often prescribed for patients with declining renal function [17,18]. However, some factors may limit the usefulness of active vitamin D or vitamin D analogue therapy, such as the resultant hypercalcemia and the inappropriately high levels of CYP24A1 activity, which, besides the deactivation of 1,25(OH)_2_D_3_, also have the potential to deactivate vitamin D analogues [19]. Considering the decline in CYP27B1 and elevated CYP24A1 activity during CKD, preventing 1,25(OH)_2_D_3_ breakdown to improve the half-life of 1,25(OH)_2_D_3_ could offer a strategy that may improve vitamin D activity and possibly improve responsiveness to vitamin D therapies [20,21].

Previously, 1,25(OH)_2_D_3_ analogue-derived specific CYP24A1 inhibition was shown to significantly lower PTH levels in normal rats without hypercalcemia [19]. Compounds with dual CYP24A1 inhibition and VDR activation were also demonstrated to ameliorate SHPT in a dose-dependent manner in uremic rats with efficient reduction of plasma PTH at doses that did not elevate serum calcium or phosphate, suggesting potential usefulness in human CKD. However, the therapeutic effect of mixed-mode analogues could not be solely attributed to the prevention of endogenous 1,25(OH)_2_D_3_ breakdown [19]. Furthermore, it has been suggested that a targeted CYP24A1 inhibition approach could be of therapeutic usefulness for other conditions associated with CYP24A1 dysregulation, such as some malignancies, psoriasis, and diabetes [19,22,23]. Recently, attention has been brought to side-chain oxime-substituted vitamin D3 analogues as potential bioactive candidates, where the synthesis of three C-24 oxime derivatives has been described without any CYP24A1 modulatory activity evaluation [22].

Among the reported synthetic analogues for calcitriol, different side-chain C-24 and C-25 oxime and oxime ether analogues have been previously described and biologically evaluated [23,24]. Recently, attention has been brought to side-chain oxime-substituted vitamin D_3_ analogues as potential bioactive candidates, where the synthesis of three C-24 oxime derivatives has been described [22]. However, none of these studies evaluated the CYP24A1 modulatory activity of the oxime analogues.

This study has established a CYP24A1 inhibitor with prospective therapeutic value for conditions showing low vitamin D activity, such as CKD. Here, we report the synthesis of a novel C-24 *O*-methyl oxime derivative of 1,25(OH)_2_D_3_ (**VD1-6**), together with an evaluation of its VDR binding affinity and its in silico docking evaluation in CYP24A1 compared to 1,25(OH)_2_D_3_. We also assess the biological effects of **VD1-6** on CYP24A1 inhibition by assessing 1,25(OH)_2_D_3_ preservation and the reduction in C-24 hydroxylation of 25(OH)D_3_ in human embryonic kidney HEK293T cells.

## 2. Materials and Methods

### 2.1. Synthesis of **VD1-6**

The reagents, chemicals, and apparatus used for the synthesis and purification of **VD1-6** are detailed in the Appendix A. The synthetic pathway followed to synthesize **VD1-6** is described in Figure 1. Nuclear magnetic resonance (NMR) and mass spectrometry (MS) spectra are provided in the Appendix A.

### 2.2. VDR Competitor Binding Assay

The PolarScreen™ VDR Competitor Assay Kit, Red (Life Technologies Australia Pty Ltd., Mulgrave, VIC, Australia) was used for the assay. VDR recombinant human protein (32,100 nM), Fluormone™ VDR Red tracer (100 nM), and dithiothreitol (DTT) solutions provided by the kit manufacturer were thawed for 30 min on ice before use. A sufficient volume of complete VDR Red screening buffer “complete buffer” (5 µL of DTT for each 1 mL of the VDR Red screening buffer) was prepared and kept on ice until use. Primary stock solutions (1 mM) of 1,25(OH)_2_D_3_, 25(OH)D_3_ (Sigma Aldrich, Castle Hill, NSW, Australia), and **VD1-6** were prepared separately in DMSO. From these solutions, 15 further 3-fold serial dilutions were prepared for each of the three compounds in DMSO, with the lowest concentration in the series of 0.07 nM. For conducting the assay, each of the 16 DMSO dilutions for each compound was further diluted 50-fold using the complete buffer for the resultant highest and lowest concentrations, being 2 × 10^4^ nM and 0.0014 nM, respectively, and for a post-dilutional DMSO concentration of 2%. To prepare the VDR/Fluormone™ complex, separate 4 nM Fluormone™ and 14.48 nM, VDR full-length protein solutions were prepared by appropriate dilution of the manufacturer’s stock solutions using complete buffer followed by vortex mixing of the Fluormone™ solution for 10 sec, and inversion mixing only of the VDR solution. Sufficient equal volumes of the Fluormone™ and VDR dilutions were combined and mixed by inversion, and the final solution (2 nM Fluormone™/7.24 nM VDR) was kept on ice until use.

In a 384 microwell plate, 10 µL aliquots from each of the 16 (0.0014–2 × 10^4^ nM) serial dilutions for each of the three tested compounds were pipetted into triplicate adjacent wells. To those wells, 10 µL of the (2 nM Fluormone™/7.24 nM VDR) solution was then added for final concentrations of test compounds ranging from 0.0007 to 1 × 10^4^ nM, final Fluormone™ concentration of 1 nM, and a final VDR full-length concentration of 3.62 nM (the manufacturer’s recommended lot-specific concentration). Maximum and minimum polarization controls were similarly prepared for monitoring assay performance by the addition of the Fluormone™/VDR complex to a column of wells with 10 µL blank 2% DMSO in complete buffer and a column of wells with 10 µL 2 × 10^4^ nM 1,25(OH)_2_D_3_, respectively. The final DMSO concentration in each well was 1%, matching the manufacturer’s recommended solvent tolerance for the assay. The plate was lid covered and shaken horizontally on an orbital shaker at 60 rpm for 5 min, then incubated for 3 h at room temperature. Fluorescence polarization values (mP) were then measured on the plate reader Cytation™ 5 multi-mode (Biotek, Winooski, VT, USA) equipped with a red fluorescence polarization filter (Excitation/Emission, 530/590 nm) at 25 °C. IC_50_ was determine by modelling of the mean fluorescence polarization readings versus corresponding concentrations for each compound, as recommended by the kit manufacturer [25]. Briefly, sigmoidal dose-response curve fitting with a variable slope was performed using GraphPad Prism^®^ 8.3.0 for Windows (GraphPad Software, San Diego, CA, USA) with the equation: Y = mP100% + (mP0% − mP100%)/[1 + 10((LogIC_50_ − X) × Hill Slope)], where: Y = mP, X = Log [inhibitor], mP100% = 100% competition, and mP0% = 0% competition.

### 2.3. CYP24A1 Molecular Docking Studies

A protein crystal structure for the human CYP24A1 enzyme has not previously been reported. As such, we utilized the crystal structure of a rat mitochondrial cytochrome P450 24A1 co-crystallized with a zwitterionic surfactant, protein data bank (PDB) ID: 3K9V (Resolution = 2.5 Å and receptor pocket-size = 2512 Å) [26]. Ligand structures for docking were acquired by determining the appropriate SMILES string for 1,25(OH)_2_D_3_ and **VD1-6**. Enumeration of 3D conformers was undertaken using OMEGA 3.3.0.3, with the default maximum number of conformers (200) [27,28]. **VD1-6** and 1,25(OH)_2_D_3_ produced 60 and 62 conformations, respectively. With the selected crystal structure for CYP24A1, binding pockets were derived using Make Receptor 3.3.0.3 [29]. Appropriate protein and co-crystallized ligand sections were chosen, and auto-generated constraints were not included, as per the default settings. No alterations to the site shape potential were made. Docking studies on the receptor were performed using FRED 3.3.0.3 as part of the OE Docking suite [30,31,32]. Docking resolution was set to “High,” and the number of poses was limited to 1. The two ligands docked successfully, and the predicted orientations were examined using VIDA 3.3.0.3 [33]. Protein–ligand interactions were further investigated using the OEChem toolkit (specifically complex2img.py). Generated Chemgauss4 scores ranked the docked 1,25(OH)_2_D_3_ and **VD1-6** according to their goodness of fit into the receptor pocket and the interactions with surrounding residues.

### 2.4. HEK293T Cell Culture Experiments

With relevance to kidney physiology, the HEK393T cell line has been sought as a common normal kidney cell line that is used to model renal physiology [34] and is known to express vitamin D responsive genes, in which adequate response to 1,25(OH)_2_D_3_ by CYP24A1 induction has been previously demonstrated [35]. HEK293T cells were maintained at 37 °C in a growth media comprised of minimum essential medium alpha (alpha-MEM), 10% *v*/*v* fetal bovine serum (FBS), and the standard usage of tissue culture additives, including 25 nM HEPES, 2 mM ʟ-glutamine, and antibiotics: 100 U/mL penicillin G/streptomycin. For experimentation, cells were seeded in 24-well tissue culture plates with 30% confluency and were allowed to attach for 24 h before treatment application.

#### 2.4.1. Assessment of Relevant Genetic Response to **VD1-6**

Upon cell attachment, culture wells (n = 5) were exposed for 24 h to vehicle control (0.2% *v*/*v* ethanol) or treatments of **VD1-6** (10^−7^ M), 1,25(OH)_2_D_3_ (10^−9^ M), or a combination of **VD1-6** (10^−11^ M or 10^−10^ M) and 1,25(OH)_2_D_3_ (10^−9^ M). The cells from replicate wells for each of the three treatment groups were then treated for RNA extraction using TRIzol^®^ reagent (Thermo-Fisher Scientific, Waltham, MA, USA) at room temperature, following the manufacturer’s protocol [36]. Briefly, after media removal, 200 µL TRIzol^®^ was added to unwashed cells in each well to lyse them. The lysate of each well was then transferred to 1.5 mL Eppendorf tubes, and 40 µL of chloroform was added; then the tube content was mixed by inverting and centrifuged for 15 min at 12,000× *g* rpm at 4 °C. The upper aqueous layer was transferred to a new tube, to which 100 µL of isopropanol was added, and the tube content was incubated for 10 min, followed by centrifugation for 10 min at 12,000× *g* rpm at 4 °C; the supernatant was then discarded. For washing, 200 µL of 70% ethanol was added to the RNA pellet, vortexed for 5 s, and centrifuged for 5 min at 7500× *g* rpm at 4 °C. The supernatant was discarded, and the RNA pellet was air-dried for 10 min. The RNA pellet was resuspended in 25 µL RNase-free water. The tubes were incubated in a 60 °C water bath for 10 min before storage at −80 °C for real-time PCR analysis.

Replicate total RNA samples were analyzed for *CYP24A1*, *CYP27B1*, *VDR*, *CALB1*, and *SLC34A3* mRNA and housekeeping gene, *ACTB*, by quantitative PCR. A total of 2µg of total RNA was reverse transcribed using the iScript™ cDNA synthesis kit (Bio-Rad, Hercules, CA, USA), following the manufacturer’s protocol [37]. A total of 12.5 ng of cDNA per sample was used for real-time RT-PCR analyses of the target gene mRNA levels using Forget-Me-Not™ EvaGreen^®^ qPCR Master Mix (Biotium, Fremont, CA, USA). Primer sequences for target mRNA were: *CYP24A1*, F 5′-CCTGCTGCCAGATTCTCTGGAA-3′, R 5′-TTGCCATACTTCTTGTGGTACTCC-3′; *CYP27B1*, F 5′-CAGACAAAGACATTCATGTGGG-3′, R 5′-GTTGATGCTCCTTTCAGGTAC-3′; *VDR*, F 5′-CCAGTTCGTGTGAATGATGG-3′, R 5′-GTCGTCCATGGTGAAGGA-3′; *CALB1*, F 5′-AGAAACTGAGGAGCTTAAGAACT-3′, R 5′-ACTCGGCTAATTTTGTGTCATCA-3′; *SLC34A3*, F 5′-ACACCTCATCGTGCAGTTGG, R 5′- AGACTGCTGTTAGTGGCGTT-3′; *ACTB*, F 5′-AAGAGATGGCCACGGCT-3′, R 5′-CAATGATCTTGATCTTCATTGTGC-3′.

#### 2.4.2. Assessment of 1,25(OH)_2_D_3_ Preservation by **VD1-6**

Cultured wells (N = 4) were exposed for 8 and 24 h to a vehicle (0.2% *v*/*v* ethanol) and **VD1-6** at 10^−7^ M as controls and to separate treatments of 1,25(OH)_2_D_3_ at 5 × 10^−10^ M, without and with **VD1-6**, at 10^−8^ M and 10^−7^ M. Supernatants from all wells were collected at 8 and 24 h post-treatment and stored with time zero samples at −80 °C for 1,25(OH)_2_D_3_ LC-MS/MS analysis.

#### 2.4.3. Assessment of 25(OH)D_3_ Catabolism Inhibition by **VD1-6**

A total of 72 h exposure of cultured wells (N = 4) was performed for similar controls, using the procedure detailed in Section 2.4.2, and for 25(OH)D_3_ at 10^−6^ M, without and with **VD1-6**, at 10^−11^, 10^−10^, 10^−9^, 10^−8^, and 10^−7^ M, each as a separate treatment. At the end of treatment, supernatants from quadruplicate wells for each group were collected and stored at −80 °C for 24,25(OH)_2_D_3_ LC-MS/MS analysis. Cells were then treated for total RNA isolation and mRNA analyses, as described in Section 2.4.1.

### 2.5. LC-MS/MS Analysis

The 24,25(OH)_2_D_3_ concentrations were analyzed in cell culture samples from Section 2.4.3, using DAPTAD derivatization and LC-MS/MS, as previously described [38]. The calibrators were prepared in phosphate-buffered saline (PBS), and the samples were appropriately diluted with PBS to bring them into the linear range of the assay. The same LC-MS/MS method was modified for 1,25(OH)_2_D_3_ analysis to enhance the sensitivity by using a higher initial sample volume (200 µL instead of 50 µL) and reconstituting the final derivatized 1,25(OH)_2_D_3_ in 20 µL and injecting 15 µL (rather than reconstituting in 25 µL and injecting 5 µL) onto the LC-MS/MS column. Multiple reaction monitoring (MRM) transitions were used for quantifying the 1,25(OH)_2_D_3_-DAPTAD derivative, and the derivative of its deuterated isotope (1,25(OH)_2_D_3_-*d_6_*), as previously reported [39]. Similar twin chromatographic peaks for *R* and *S* isomers of the derivatives [39] (Appendix A) were also obtained and used similarly for quantification. The linearity, accuracy, and precision data for 1,25(OH)_2_D_3_ LC-MS/MS analysis are provided in Appendix A.

### 2.6. Statistical Analysis

Statistical analysis was performed using one-way analysis of variance (ANOVA) for non-parametric data, along with a secondary Tukey’s multiple comparisons test to determine the difference between treatment groups. A *p*-value less than 0.05 was considered statistically significant.

## 3. Results

### 3.1. Chemistry

**VD1-6** was synthesized as outlined in Figure 1. Iodo **VD1-1**, which was accessed from vitamin D_2_ following previously reported methods [40,41], was reacted with an enolate formed in situ from 3,3-dimethyl-2-butanone, to give ketone **VD1-2** in 72% yield. The *O*-methyloxime functionality was installed using methoxyamine hydrochloride in pyridine to give **VD1-3** in 69%. Desilylation for 1 h at room temperature using camphor sulphonic acid (CSA) revealed the secondary alcohol, which was then oxidized using pyridinium chromate (PDC) to give ketone **VD1-4** in a 68% yield over the two steps. Horner–Wadsworth–Emmons (HWE) chemistry was then employed to react ketone **VD1-4** with commercially available phosphonate **VD1-5**, under basic conditions, to conjugate the two fragments by an olefin linker. Removal of the silyl protecting groups, again using CSA, revealed the anti-planar hydroxy groups in **VD1-6** in a 43% yield over two steps.

### 3.2. Binding of **VD1-6** to VDR

Results of VDR competitor assay experiments for **VD1-6**, 1,25(OH)_2_D_3_, and 25(OH)D_3_ are represented in Figure 1. Curve fitting (solid connecting lines, Figure 1) using a sigmoidal dose–response (variable slope) curve yielded modelled binding IC_50_ values of 0.93, 56.2, and 438 nM for 1,25(OH)_2_D_3_, 25(OH)D_3_ and **VD1-6**, respectively. These concentrations are required to displace half of the high affinity fluorescent tracer from its VDR binding, resulting in a corresponding drop in the mean fluorescence polarization by 50%. This illustrates that **VD1-6** has approximately 470-fold (438/0.93) lower binding affinity than 1,25(OH)_2_D_3_ to the active site of VDR.

### 3.3. Comparative In Silico Docking of 1,25(OH)_2_D_3_ and **VD1-6** into CYP24A1

The ligand-binding site demonstrates the prominence of similar amino acid residues to those indicated by Jayaraj et al. [42], i.e., GLU 329 and THR 330 for the CYP24A1 active binding tunnel (Figure 2). Docking studies revealed that 1,25(OH)_2_D_3_ is predicted to have one hydrogen bond between its terminal C-25 hydroxyl group and THR 330. At the same time, **VD1-6** displays a single hydrogen bond to MET 246 from the C-1 hydroxyl group (Figure 2). The predicted docking pose of 1,25(OH)_2_D_3_ is consistent with the results of Annalora et al. [26], where the C-1 and C-3 hydroxyl groups are located in the vicinity of the THR 395 residue, while **VD1-6** displayed an inverse orientation to that of 1,25(OH)_2_D_3_, as shown in Figure 2. Both molecules have suitable flexibility in their structure and availability of space within the binding pocket to interact with the CYP24A1 heme core at their respective C21-25 ends. Docking scores revealed **VD1-6** to rank higher than the endogenous ligand 1,25(OH)_2_D_3_ (−12.30 and −11.64 relative Chemgauss4 docking scores) in their docking to the CYP24A1 active binding site, which indicates a higher relative predicted binding affinity of **VD1-6** to CYP24A1.

### 3.4. Effect of **VD1-6** on CYP24A1, CYP27B1, and VDR Genes

A comparative study of the effects of **VD1-6** versus 1,25(OH)_2_D_3_, alone and in combination, on the mRNA levels of *CYP24A1*, *CYP27B1*, *VDR*, *CALB1*, and *SLC34A3* in HEK293T cells at 24 h is illustrated in Figure 3. At 10^−7^ M, **VD1-6** did not affect *CYP24A1* mRNA levels, compared to vehicle control. However, 1,25(OH)_2_D_3_ at 10^−9^ M concentration was associated with a significant (*p* < 0.05) elevation in *CYP24A1* and CalB1 mRNA levels. The combination of **VD1-6** (10^−10^ M) and 1,25(OH)_2_D_3_ (10^−9^ M) resulted in a 2-fold increase in *CYP24A1* mRNA (*p* < 0.05) and a 1.3-fold increase in *CALB1* mRNA, when compared to levels with 1,25(OH)_2_D_3_ treatment alone. In contrast, mRNA levels for *CYP27B1*, *VDR*, and *SLC34A3* were not elevated by 1,25(OH)_2_D_3_, with or without **VD1-6**. Furthermore, 1,25(OH)_2_D_3_, in combination with **VD1-6** at 10^−11^ M and 10^−10^ M, decreased *VDR* mRNA levels when compared to vehicle control levels.

### 3.5. Effect of **VD1-6** on Preserving 1,25(OH)_2_D_3_ in HEK293T Cell Culture

Results of LC-MS/MS analysis of the concentration of the added 1,25(OH)_2_D_3_ to HEK293T cell culture, in the absence and presence of **VD1-6**, at concentrations of 10^−8^ and 10^−7^ M at time zero and 8 h and 24 h post-treatment, are illustrated in Figure 4. In the absence of **VD1-6**, 1,25(OH)_2_D_3_ concentrations exhibited a rapid mean decline of 83 pM (*p* < 0.05) over 8 h to reach a mean concentration of 451 ± 27.7 pM. This decline rate was found to slow beyond 8 h (i.e., 1st order rate) with a further, but non-significant mean decline of 29 pM over the following 16 h. This resulted in a mean of 1,25(OH)_2_D_3_ levels of 422 ± 11.9 pM at 24 h, which was significantly different from the time zero mean levels (*p* < 0.05) (Figure 4).

The inclusion of **VD1-6** resulted in the dose-dependent preservation of the added 1,25(OH)_2_D_3_ levels. **VD1-6** at 10^−8^ M did not exert a significant inhibition of catabolism at 8 h in terms of mean 1,25(OH)_2_D_3_ levels, compared to levels in the absence of **VD1-6** at a similar time. However, with that **VD1-6** concentration, the level of catabolism inhibition seen at 8 h seemed to be sustained up to 24 h, which resulted in a mean 1,25(OH)_2_D_3_ concentration of 480 ± 28.3 pM, which was significantly different (*p* < 0.05) when compared to that at 24 h without **VD1-6** (Figure 4). With a higher **VD1-6** concentration though, i.e., 10^−7^ M, significant catabolism inhibition was evident by 8 h, with mean 1,25(OH)_2_D_3_ concentrations of 534 ± 25.9 pM, which was significantly different (*p* < 0.05) from the mean concentration in the absence of **VD1-6** at 8 h and non-significantly different from the mean concentrations at time zero (Figure 4). This inhibition of the catabolism effect persisted up to 24 h (i.e., non-significantly different from that at 8 h and zero time points), with a mean 1,25(OH)_2_D_3_ concentration of 506 ± 28.9 pM, which was significantly different (*p* < 0.05) from mean concentrations at 24 h in the absence of **VD1-6** (Figure 4). It is of note that at 24 h, mean 1,25(OH)_2_D_3_ concentrations with both 10^−8^ M and 10^−7^ M **VD1-6** were not significantly different, possibly indicating the comparable longer-term preservation effects of these two concentrations.

### 3.6. Effect of **VD1-6** on 24,25(OH)_2_D_3_ Production in HEK293T Cell Culture

To assess the effects of **VD1-6** on 25(OH)D_3_ catabolism, HEK293T cells were treated with 25(OH)D_3_ (10^−6^ M) over 72 h, and media levels of 24,25(OH)_2_D_3_ and mRNA levels for *CYP24A1* were measured in the absence or presence of **VD1-6** (10^−11^–10^−7^ M) (Figure 5A,B). In the presence of 25(OH)D_3_ (10^−6^ M), HEK293T cells produced 24,25(OH)_2_D_3_, which was approximately 170-fold higher than the levels in the vehicle-treated cells (Figure 5A). The inclusion of **VD1-6** resulted in a biphasic concentration-dependent effect on the mean accumulated 24,25(OH)_2_D_3_ concentrations. At 10^−10^ M **VD1-6**, a 14% increase in mean 24,25(OH)_2_D_3_ concentration was observed when compared to levels with 25(OH)D_3_ treatment alone (*p* < 0.05) (Figure 5A). However, at 10^−7^ M **VD1-6**, a marked decrease in 24,25(OH)_2_D_3_ levels was seen when compared to all other treatment concentrations (Figure 5A). Although 25(OH)D_3_ treatment increased *CYP24A1* mRNA levels in HEK293T cells, **VD1-6** inclusion did not elevate *CYP24A1* mRNA levels (Figure 5B). While the elevated 24,25(OH)_2_D_3_ levels at 10^−10^ M **VD1-6** appeared to correlate with elevated *CYP24A1* mRNA levels, this was not statistically significant. The decline in 24,25(OH)_2_D_3_ levels in 25(OH)D_3_ treatments with 10^−7^ M **VD1-6** did not correspond to a change in *CYP24A1* mRNA levels when compared to the mRNA levels in 25(OH)D_3_ only treatments, suggesting that the 24,25(OH)_2_D_3_ decline was not due to reduced CYP24A1 expression.

## 4. Discussion

In this study, we have synthesized and evaluated a novel C-24 *O*-methyl oxime derivative of vitamin D_3_ for its use as a CYP24A1 inhibitor. Since it was previously shown that the absence of the terminal hydroxyl group of the 1,25(OH)_2_D_3_ side chain is not an indication that VDR binding would be abolished, it was essential to establish whether **VD1-6** has an affinity for VDR binding at concentrations used in the biological assessment. We have conducted an in vitro competition VDR binding assay based on the variable displacement of Fluormone™, a high-affinity VDR-binding fluorescent tracer [43]. This assay demonstrated that **VD1-6** has a 470-fold lower affinity to the VDR binding pocket than 1,25(OH)_2_D_3_. Furthermore, **VD1-6** binding to VDR was only evident at concentrations greater than 10^−7^ M (Figure 1). The 25(OH)D_3_, known to bind VDR with markedly less affinity than 1,25(OH)_2_D_3_ [44], had an 8-fold higher affinity for VDR than **VD1-6**.

The computer-aided simulated measurement of the change of molecular free energy that occurs due to binding (relative binding free energy) is considered a potentially reliable measure for predicting binding affinities of small ligand molecules to protein targets [45,46]. In silico docking of **VD1-6** into the CYP24A1 binding pocket was conducted as a predictive assessment of CYP24A1 binding site affinity compared to 1,25(OH)_2_D_3_. The lack of a crystal structure of the human CYP24A1 in the Protein Data Bank has led us to utilize the available rat CYP24A1 protein structure for our docking studies. The reported similarity between human and rat CYP24A1 is 85%, with the rat variant matching 11 of 13 key amino acids for substrate binding and catalysis [26,42]. Simulated docking into the active binding pocket of CYP24A1 has shown that the binding of **VD1-6** involves a change in the hydrogen bonding pattern relative to 1,25(OH)_2_D_3_. This may be attributed to the lack of a terminal hydroxyl moiety at the C-25 end of **VD1-6**, which may also explain the predicted inversion of orientation inside the binding pocket, as seen in Figure 2. The docking score of **VD1-6** was superior to that of 1,25(OH)_2_D_3_, reflected by the lower relative binding free energy of −12.30 compared to −11.64 of 1,25(OH)_2_D_3_, which indicates a higher simulated binding affinity of **VD1-6** to CYP24A1.

It is noteworthy that for 1,25(OH)_2_D_3_ to be hydroxylated at either C-23 or C-24 in the active site of CYP24A1, the terminal side chain (C-21 to C-25) needs to be positioned towards the heme moiety of the enzyme [47]. Our simulated poses for 1,25(OH)_2_D_3_ and **VD1-6** appear consistent with this concept (Figure 2), with aliphatic side chains pitched toward the heme moiety, with appropriate flexibility. From a metabolism perspective, the C-24 *O*-methyl oxime moiety of **VD1-6** would block C-24 hydroxylation, making it only possible to proceed down the C-23 pathway. On the other hand, the endogenous ligand 1,25(OH)_2_D_3_ can proceed through both the C-23 and C-24 catabolic routes. The greater potential for the interaction of **VD1-6** with CYP24A1, besides its occupied C-24, suggests that **VD1-6** is a more efficient CYP24A1 binder, with partial catabolic immunity, i.e., a likely inhibitor of vitamin D catabolism, with a presumably longer half-life than the endogenous ligand.

The combined in vitro cell-free VDR binding assay and in silico docking results for **VD1-6**, in comparison to the endogenous ligand, provide a clear preliminary indication that **VD1-6** has the potential to be an inhibitor of CYP24A1, without significant VDR binding, at least up to concentrations of 10^−7^ M. Consistent with this, **VD1-6** alone at 10^−7^ M did not induce *CYP24A1* mRNA, unlike 1,25(OH)_2_D_3_, indicating the absence of VDR binding by **VD1-6** (Figure 3). However, the inclusion of **VD1-6**, at lower concentrations, with 1,25(OH)_2_D_3_ treatment increased mRNA levels for *CYP24A1* and *CALB1* when compared to the effects of 1,25(OH)_2_D_3_ treatment alone. This suggests that **VD1-6** potentiated the transcriptional effects of 1,25(OH)_2_D_3_, likely through binding to and inhibiting CYP24A1, increasing the half-life of 1,25(OH)_2_D_3_. The gene CALB1 encodes for Calbindin-D28K, and like CYP24A1, is directly induced by 1,25(OH)_2_D_3_ [48]. In contrast, **VD1-6** did not impact *CYP27B1* mRNA levels, suggesting that the effects of **VD1-6** did not alter 1,25(OH)_2_D_3_ synthesis. *VDR* mRNA levels declined by 2-fold with the addition of **VD1-6** and 1,25(OH)_2_D_3_. While the gene for VDR is vitamin D-responsive, previous studies have also demonstrated that 1,25(OH)_2_D_3_ binding VDR can increase the half-life of VDR protein without inducing *VDR* expression [49]. Further studies would be required to establish whether an increased half-life of 1,25(OH)_2_D_3_ due to **VD1-6** treatment results in an increased VDR half-life. The 1,25(OH)_2_D_3_ mRNA levels were also unchanged due to **VD1-6** treatment. *SCL34A3*, encoding for proximal tubular sodium-phosphate cotransporter, NaPi-IIc, has been shown to be stimulated by 1,25(OH)_2_D_3_ [50]. However, the effects of 1,25(OH)_2_D_3_ on NaPi-IIc have also been shown to be secondary to the regulation by the phosphaturic hormone, FGF23 [50]. While CALB1 mRNA levels were elevated in the presence of 1,25(OH)_2_D_3_ and **VD1-6**, it is worth noting that the elevation of *CYP24A1* mRNA may contribute to the effects of 1,25(OH)_2_D_3_ and **VD1-6** on gene expression in this experiment by responding to an increased half-life of 1,25(OH)_2_D_3_ by increasing its catabolic activity, potentially contributing to an absence of mRNA increases for *VDR* and *SCL34A3*.

To establish whether reduced 1,25(OH)_2_D_3_ catabolism occurred with **VD1-6** treatment, we tested the preservation of 1,25(OH)_2_D_3_ in HEK293T cell cultures up to 24 h (Figure 4). While 1,25(OH)_2_D_3_ levels declined by 16% at 8 h and 22% at 24 h, **VD1-6** at 10^−7^ M prevented this decline (Figure 4). While **VD1-6** at 10^−8^ M was less protective of 1,25(OH)_2_D_3_ levels at 8 h, at 24 h, protection was comparable to that of 10^−7^ M **VD1-6** (Figure 4). **VD1-6** at 10^−7^ M was also capable of inhibiting 25(OH)D_3_ catabolism, as measured by the reduced formation of 24,25(OH)_2_D_3_, the first product of 25(OH)D_3_ catabolism (Figure 5A), most likely through significant competition and occupation of the CYP24A1 active site. Interestingly, the elevation in 24,25(OH)_2_D_3_ levels with **VD1-6** at 10^−10^ M may have occurred due to counter-elevation of CYP24A1 activity by the preservation of 1,25(OH)_2_D_3_ (Figure 5A,B), given that **VD1-6** combined with 1,25(OH)_2_D_3_ enhanced *CYP24A1* mRNA expression (Figure 3).

## 5. Conclusions

Herein, we have successfully synthesized a novel C-24 *O*-methyloxime analogue of vitamin D_3_ and evaluated its VDR and CYP24A1 interactions and its CYP24A1 inhibitory activity. **VD1-6** has negligible VDR binding activity and is predicted to bind, with high affinity, to the CYP24A1 active pocket, thus blocking CYP24A1 catabolic activity. Thus, our findings indicate that this compound is a novel pure inhibitor of CYP24A1. Future work will investigate the capacity for **VD1-6** to inhibit CYP24A1 in diseases marked by low vitamin D activity and excessive vitamin D catabolism.

## Data Availability

The data presented in this study are available on request from the corresponding author. The data are not publicly available, as part of the data will be used for further studies.

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
