# Peer review of "Therapeutic Potential of a Novel Vitamin D3 Oxime Analogue, VD1-6, with CYP24A1 Enzyme Inhibitory Activity and Negligible Vitamin D Receptor Binding"

_biomolecules, 2022, doi:10.3390/biom12070960_

Round 1

Reviewer 1 Report

This manuscript by Alshabrawy et al. describes the synthesis and study of the biological effects of a novel vitamin D3 oxime analog denominate VD1-6.

 The investigators found that VD1-6 has poor VDR binding activity, and binds in-silico with high affinity to the CYP24A1 active pocket, thus blocking CYP24A1 catabolic activity. The authors suggest that VDR1-6 could be a potential candidate for treating some disorders, such as secondary hyperparathyroidism.

There are some interesting results, for example, docking studies to explain the inhibitory properties of the new analog. However, many typographical errors are detected and the authors need to clarify the next question:

-Why do the authors not evaluate calcium levels in vivo? This analog could increase the levels of 1,25 (OH)2 D3 levels and therefore induce hypercalcaemia.

 Minor points:

a) Figure 1, 2, 4 and 5 are duplicated in the manuscript. I guess it will be a mistake. 

b) 3.1 chemistry

L1 replace Iodo VD1-1 by iodide VD1-1

L3 replace 3, 3–diemthyl-2-butanone by 3,3-dimethyl-2-butanone

L7 Replace Horner-Wadsworth-Emmons (HWE) chemistry by Wittig-Horner (WH) chemistry

L11 check 43%!!  yield for the two steps 

-          SCHEME1

Stereochemistries at C14 are missing 

c) Supplementary data:

PAGE 2

Synthesis of VD1-2

L3      3,3-dimethyl-2-butanone (0.37 mL  ≠ 2.10 mmol)

L5         VD1-1 (130 mg      0.30 mmol)   

PAGE 3

Synthesis of VD1-3:

L1   Replace VD1-4 by VD1-2      

L1  300 mg ≠  0.74 mmol

Synthesis of VD1-4:

L1 Replace VD1-4 by VD1-3

L1 220 mg ≠ 0.50 mmol 

PAGE 4

Synthesis of VD1-6

L9  CSA ( 60 mg  ≠ 0.32 mmol)

L13  30 mg ≠  43% steps). It should be 13.4%

HRMS are missing for the new compounds.

Reviewer 2 Report

The authors synthesized a novel analogue of vitamin D3 (VD1-6) that contain a oxyme-substituted side chain at C-24, and demonstrated the moderate affinity for VDR. In contrast, the authors predict high affinity for CYP24A1 by in silico docking analysis and also demonstrated that VD1-6 have the potential to be CYP24A1 specific inhibitor via HEK293T cell assays. It is interesting that VD1-6, despite being an analog of vitamin D3, has a reduced affinity for VDR and has acquired high affinity and inhibitory activity for CYP24A1. This is also an important finding for development therapeutic candidate for such as SHPT.

Therefore, the manuscript is recommended for publication after some minor revisions as below, particularly corrections to the wrong figure display.

1.     Figures 1, 4, and 5 have duplicated panels. Incorrect display of Figure 2 at page 12.

2.     There are some typos such as 
page 2, "
 However, the therapeutic effect of mixed-mode compounds[19]. However, …"
page 3, "ThisThis study has established …"
page 12, The second sentence in Discussion section is difficult to understand. It should be rewritten as a concise expression.
page 13, "… in silico docking results for- forfor VD1-6 …"

3.     Figure 2   It seems that the binding pocket of CYP24A1 is extremely large for the ligand molecules. The authors pointed out only one hydrogen bond between Met246 and C-1 hydroxy, but it is unlikely that this alone will provide high affinity of VD1-6. Also, VD1-6 was bound in the opposite direction compared to that of 1,25(OH)2D3. How is oxime at C-24 related to this? Is there any hydrogen bond?

The authors used the rat CYP24A1 model for docking analysis because there is no structural information for human CYP24A1. I would suggest to use a homology model or AlphaFold model of human CYP24A1.

Reviewer 3 Report

Major comments:

1. Please provide more details how dose response curses were fitted and IC50 values were determined, i.e., sigmoidal dose-response modelling with varying slopes should be explained in the method section.

2. Fig. 2 is suboptimal in its presentation, please take more effort in displaying the binding cleft of the enzyme and the involved amino acids. Moreover, the figure is displayed twice, please take more care in manuscript preparation.

3. More strong vitamin D target genes should be tested for possible regulatory effects of VD1-6, ideally in appropriated tissues or cell lines.

Minor comments:

1.  Please check again the whole manuscript for typos, such as arere" and "contributingcontributing".

2. All abbreviations need to be defined at their first time use and the applied consistently, e.g., NMR and MS.

3. Details od suppliers need to be defined only at first time, international suppliers, such as Sigma, maybe not at all.

Round 2

Reviewer 3 Report

The manuscript improved, but there are still open issues:

1. The answer my point 3 (further vitamin D target genes) is unsatisfactory. At least 3 strong vitamin D target genes should be investigated.

2. Fig. 4 takes too much space in the manuscript in relation to other more important figures.

3. The cited references are not representing the field, i.e. the list is a bit biased.
